# Stable Gastric Pentadecapeptide BPC 157 Therapy of Rat Glaucoma

**DOI:** 10.3390/biomedicines10010089

**Published:** 2021-12-31

**Authors:** Tamara Kralj, Antonio Kokot, Mirna Zlatar, Sanja Masnec, Katarina Kasnik Kovac, Marija Milkovic Perisa, Lovorka Batelja Vuletic, Ana Giljanovic, Sanja Strbe, Suncana Sikiric, Slaven Balog, Bojan Sontacchi, Dijana Sontacchi, Matko Buljan, Eva Lovric, Alenka Boban Blagaic, Anita Skrtic, Sven Seiwerth, Predrag Sikiric

**Affiliations:** 1Department of Pharmacology, School of Medicine, University of Zagreb, 10000 Zagreb, Croatia; tamara_kralj@yahoo.com (T.K.); mirnazlatar@yahoo.com (M.Z.); sanjamp@yahoo.com (S.M.); kkasnik@gmail.com (K.K.K.); giljanovic.ana.hr@gmail.com (A.G.); strbes@gmail.com (S.S.); mbuljan@kbd.hr (M.B.); abblagaic@mef.hr (A.B.B.); 2Department of Anatomy and Neuroscience, Faculty of Medicine, J.J. Strossmayer University of Osijek, 31000 Osijek, Croatia; 3Department of Pathology, School of Medicine, University of Zagreb, 10000 Zagreb, Croatia; marija.milkovic@mef.hr (M.M.P.); lbatelja@mef.hr (L.B.V.); suncanasikiric@gmail.com (S.S.); eva.lovric@kb-merkur.hr (E.L.); skrtic.anita@gmail.com (A.S.); sven.seiwerth@mef.hr (S.S.); 4Ophthalmology Clinic Dr. Balog, 31000 Osijek, Croatia; slavenbalog@gmail.com; 5General Hospital Cakovec, 40000 Cakovec, Croatia; bojan_sontacchi@yahoo.com (B.S.); dijana.sontacchi@yahoo.com (D.S.)

**Keywords:** pentadecapeptide BPC 157, glaucoma, rat, therapy

## Abstract

Cauterization of three episcleral veins (open-angle glaucoma model) induces venous congestion and increases intraocular pressure in rats. If not upgraded, one episcleral vein is regularly unable to acquire and take over the whole function, and glaucoma-like features persist. Recently, the rapid upgrading of the collateral pathways by a stable gastric pentadecapeptide BPC 157 has cured many severe syndromes induced by permanent occlusion of major vessels, veins and/or arteries, peripherally and centrally. In a six-week study, medication was given prophylactically (immediately before glaucoma surgery, i.e., three episcleral veins cauterization) or as curative treatment (starting at 24 h after glaucoma surgery). The daily regimen of BPC 157 (0.4 µg/eye, 0.4 ng/eye; 10 µg/kg, 10 ng/kg) was administered locally as drops in each eye, intraperitoneally (last application at 24 h before sacrifice) or per-orally in drinking water (0.16 µg/mL, 0.16 ng/mL, 12 mL/rat until the sacrifice, first application being intragastric). Consequently, all BPC 157 regimens immediately normalized intraocular pressure. BPC 157-treated rats exhibited normal pupil diameter, microscopically well-preserved ganglion cells and optic nerve presentation, normal fundus presentation, normal retinal and choroidal blood vessel presentation and normal optic nerve presentation. As leading symptoms, increased intraocular pressure and mydriasis, as well as degeneration of retinal ganglion cells, optic nerve head excavation and reduction in optic nerve thickness, generalized severe irregularity of retinal vessels, faint presentation of choroidal vessels and severe optic nerve disc atrophy were all counteracted. In conclusion, we claim that the reversal of the episcleral veins cauterization glaucoma appeared as a consequence of the BPC 157 therapy of the vessel occlusion-induced perilous syndrome.

## 1. Introduction

Cauterization of three episcleral veins (open-angle glaucoma model) induces venous congestion and increases intraocular pressure in rats [1,2]. If not upgraded, one episcleral vein is regularly unable to acquire and take over the whole function, and glaucoma-like features persist [1,2]. As a novel therapeutic approach, we investigated the stable gastric pentadecapeptide BPC 157 therapy (for review, see [3,4,5,6]).

Recently, the rapid upgrading of the collateral pathways by a stable gastric pentadecapeptide BPC 157 has cured many severe syndromes [7,8,9,10,11,12,13,14,15,16] induced by permanent occlusion of major vessels, veins and/or arteries [7,8,9,10,11,12,13], peripherally and centrally, major intoxication (alcohol, lithium) [14,15] and maintained intra-abdominal hypertension [16]. Its applicability in the rapid upgrading of the collateral pathways may likely provide an additional beneficial effect in various vessel tributaries and normalization/attenuation of the intracranial (sinus sagittal) hypertension, portal and caval hypertension and aortal hypotension, and counteraction of the multiorgan failure syndrome [7,8,9,10,11,12,13,14,15,16]. Consequently, we theorized that all BPC 157 regimens [7,8,9,10,11,12,13,14,15,16] in rats with cauterized three episcleral veins and increased intraocular pressure would immediately normalize intraocular pressure.

Thereby, through BPC 157 therapy, both as a rapid effect and then sustained beneficial effect, one remaining episcleral vein should be upgraded to compensate otherwise inescapable venous congestion and counteract and avoid the increased intraocular pressure and consequent injurious course, thus preventing otherwise rapidly imminent glaucoma course and an advanced increased intraocular course reversal.

Furthermore, taking retinal ischemia as the final drawback [1,2], the nitric oxide synthase (NOS)-blocker L-NAME retrobulbar administration-induced retinal ischemia in rats was counteracted by BPC 157 application [17]. This was likely due to its modulatory effects on NO system [18] and prostaglandins system [19], maintained vasomotor tone through the activation of Src-Caveolin-1-eNOS pathway [20] and interaction with several molecular pathways [20,21,22,23,24,25,26,27,28,29]. When administered during reperfusion in the stroke rats [21], BPC 157 therapy counteracted both early and delayed neural hippocampal damage and completely recovered debilitated functions in the stroke rats [21]. There is also the maintenance of the function of thrombocytes (without interference with coagulation) [30,31,32] and its action as a stabilizer of cellular junctions [25] and a free radical scavenger [33,34,35], in particular in vascular occlusion studies [7,8,9,10,11,12,36,37,38,39,40].

Such beneficial effect may have a more general significance regarding the eye. This may appear along with the maintained corneal transparency [41,42] after healing of the perforative corneal injury [41] and total debridement of corneal epithelium [42]. BPC 157 counteracted the damaging effects of lacrimal gland extirpation and dry eye syndrome in rats [43]. In guinea pigs and rats, we also revealed a NO system dependence of the atropine-induced mydriasis and L-NAME- and L-arginine-induced miosis and reversal by the pentadecapeptide BPC 157 in both rats and guinea pigs [44].

Thus, we hypothesized that leaving the one remaining episcleral vein as a possible rescuing pathway [45] and BPC 157 therapy would absorb the brunt of the cauterization of three episcleral veins in glaucomatous rats. In addition, we used both injured eyes to provide more severe and more reliable glaucomatous injury, as primary open-angle glaucoma is typically a bilateral progressive chronic optic neuropathy [46,47,48]. This would skip the rule of one eye being used as experimental and the other as control. Thereby, we would cause less injury in glaucomatous rats, incompletely defined, since contralateral retinas in unilateral models of retinal injury show neuronal degeneration and glial activation as well [49]. Probably not only in theory, BPC 157 therapy in the form of eye drops, intraperitoneal or per-oral, in drinking water, prophylactic as well as curative, would complement the regular anti-glaucoma therapy options, as well as laser trabeculoplasty, and thereby avoid their possible complications and adverse effects [50,51].

## 2. Materials and Methods

### 2.1. Animals

This study was conducted with 12-week-old, 200 g body weight, male albino Wistar rats, randomly assigned at 6 rats/group/interval. The rats were bred in-house at the Pharmacology Animal Facility, School of Medicine, Zagreb, Croatia. The animal facility was registered by the Directorate of Veterinary (Reg. No: HR-POK-007). Laboratory rats were acclimated for five days and randomly assigned to their respective treatment groups. Laboratory animals were housed in polycarbonate (PC) cages under conventional laboratory conditions at 20–24 °C, relative humidity of 40–70% and noise level 60 dB. Each cage was identified with dates, study number, group, dose, number and sex of each animal. Fluorescent lighting provided illumination 12 h per day. Standard good laboratory practice (GLP) diet and fresh water were provided ad libitum. Animal care was in compliance with standard operating procedures (SOPs) of the Pharmacology Animal Facility, and the European Convention for the Protection of Vertebrate Animals was used for Experimental and other Scientific Purposes (ETS 123).

This study was approved by the local Ethics Committee. Ethical principles of the study complied with the European Directive 010/63/E, the Law on Amendments to the Animal Protection Act (Official Gazette 37/13), the Animal Protection Act (Official Gazette 135/06), the Ordinance on the protection of animals used for scientific purposes (Official Gazette 55/13), Federation of European Laboratory Animal Science Associations (FELASA) recommendations and the recommendations of the Ethics Committee of the School of Medicine, University of Zagreb. The experiments were assessed by observers blinded as to the treatment.

### 2.2. Drugs

Medication was administered as described previously [7,8,9,10,11,12,13,14,15,16], without use of a carrier or peptidase inhibitor, for stable gastric pentadecapeptide BPC 157, a partial sequence of the human gastric juice protein BPC, which was freely soluble in water at pH 7.0 and in saline. BPC 157 (GEPPPGKPADDAGLV, molecular weight 1419; Diagen, Ljubljana, Slovenia) was prepared as a peptide with 99% high-performance liquid chromatography (HPLC) purity, with 1-des-Gly peptide being the main impurity. The dose and application regimens were as described previously [7,8,9,10,11,12,13,14,15,16].

Thiopental (Rotexmedica, Trittau, Germany), diazepam (Apaurin, Krka, Novo Mesto, Slovenia), tropicamide (Mydriacyl 1% Alcon, Camberley, UK) (pupil dilatation), and tetracaine (Tetracaine, Pliva, Zagreb, Croatia) were also used.

### 2.3. Experimental Protocol

#### 2.3.1. Episcleral Veins Cauterization

Briefly, in deeply anesthetized rats (40 mg/kg thiopental and 10 mg/kg diazepam, given intraperitoneally), two drops in each eye of tropicamide (Mydriacyl 1% Alcon, Camberley, UK) (pupil dilatation) and tetracaine (Tetracaine, Pliva, Zagreb, Croatia) (local anesthesia) were administered. Two dorsal episcleral veins and one temporal episcleral vein were isolated from the surrounding tissues, as described [45]. A cautery was specifically applied to the selected vein with extra caution to spare surrounding tissue.

#### 2.3.2. Medication

Medication started prophylactically (immediately before glaucoma surgery) or as curative treatment (at 24 h after glaucoma surgery). Medication included BPC 157 daily regimen (0.4 µg/eye, 0.4 ng/eye; 10 µg/kg, 10 ng/kg) given locally as eye drops in each eye, intraperitoneally (last application at 24 h before sacrifice) or per-orally in drinking water (0.16 µg/mL, 0.16 ng/mL, 12 mL/rat until the sacrifice, first application given intragastrically). Controls simultaneously received an equal volume of distilled water (two drops/eye), saline intraperitoneal application (5 mL/kg) or per-oral drinking water (12 mL/day/rat).

#### 2.3.3. Intraocular Pressure, Pupillary Function, Fundoscopy and Histopathology of the Retina and Optic Nerve Head

##### Intraocular Pressure

As described before [17], intraocular pressure was measured at 7.00 p.m. using a calibrated applanation tonometer Tonopen XL by Reichert Technologies (the transducer was placed in the same spot of the cornea for each measurement). The assessment points were at 1 min and 5 min after first initial application, then at 24 h thereafter, as well as at the end of the fourth and sixth post-injury week. The rats were deeply anesthetized with diazepam (5 mg/kg b.w. i.p.) and sodium thiopental (5 mg/kg b.w. i.p.), and furthermore, tetracaine drops were administered in both eyes. Intraocular pressure was assessed in normal rats, and values between 13 and 19 mmHg were considered normal, in accordance with previous rat data [17,41].

##### Pupillary Function

As described before [41], immediately prior to and after treatment, the eyes were photographed. The assessment points were at 1 min and 5 min after first initial application, then at 24 h thereafter, as well as at the end of the fourth and sixth post-injury week. The eyes were imaged on a Veho Discovery VMS-004 Deluxe USB microscope camera with its own light source. Each photograph took 2–3 s under medium-bright light (light intensity set to 50% capability of the camera) at 0.5 cm from the eye. The pictures were processed and analyzed with the accompanying software to calculate the diameter, area and circumference of the pupils. Prior to each recording, the camera was calibrated using graph paper. Based on our assessment, a pupil radius of 0.3–0.5 mm in rats was considered to be the normal range [41].

##### Fundoscopy

Posterior eye segment assessment, and thus vascularization of the eye fundus and presentation of the optic nerve head, were filmed with USB microscope camera Veho Discovery VMS-004 Deluxe and VOLK Digital Wide Field Lens for indirect ophthalmoscopy at 24 h, 4 weeks and 6 weeks after procedure in rats that underwent prophylactic regimen or in rats that received curative regimen (at 24 h after glaucoma surgery), before initial application, and after therapy at 1 h and 24 h, as well as at the end of the fourth and sixth post-injury week. We focused our images of the vessels, extending into and out of the optic disc. Retinal changes/alterations in vessel caliber and tortuosity were recorded, and optic disc pallor was observed and scored (score 0–3). The degree of ischemia of the retina was scored 0–3 by taking images of the animals’ fundus as follows: 0: orderly eye background, normal presentation of the choroidal blood vessels (normal reddish fundus background color); 1: discrete generalized irregularity in the diameter of the blood vessels with mild atrophy of the optic disc, bright presentation of the choroidal blood vessels (bright reddish fundus background color); 2: moderate generalized irregularity of the diameter of blood vessels with moderate atrophy of the optic disc, barely visible (faint presentation) choroidal blood vessels (brighter fundus background color); and 3: strong generalized irregularity of the diameter of the blood vessels with severe atrophy of the optic disc, barely visible (extremely faint presentation) choroidal blood vessels (bright fundus background color). During recording, the animals received artificial tears (Isopto Tears, Alcon Pharmaceuticals, Surrey, UK) in the eyes. Moistening of the cornea allows better visualization of the fundus and clearer images of the retina. The images were processed with software purchased with the USB microscope camera Veho Discovery VMS-004 Deluxe.

##### Histopathology of the Retina and Optic Nerve Head

In the histopathological evaluation at the end of 24 h, 4 weeks and 6 weeks, the enucleated eyes were fixed in 4% phosphate-buffered formalin, and transverse sections passing through the optic nerve were obtained. The specimens were processed on paraffin wax. Paraffin sections of 5-μm thickness were obtained and stained with hematoxylin and eosin for light microscopy.

The thickness of the retinal tissue, ganglion cell layer, inner nuclear layer, outer nuclear layer, optic nerve and anterior prelaminar region of optic nerve head was analyzed and measured with software ISSA program (VamsTec, Zagreb, Croatia), using the area of maximal tissue damage detected in semi-serial slide sections. A total of five high-power fields were randomly selected for analysis, conducted with an ocular micrometer at ×200 magnification within 0.5 mm from the optic disc. Histopathologic changes, such as edema, vacuolar degeneration and pyknosis, as well as neutrophilic leucocyte infiltration, were noted if present. The histological examination was performed by a pathologist, who was unaware of the treatment.

### 2.4. Statistical Analysis

We used Statistica 12.1. for Windows to perform statistical analysis. Distribution of data normality was tested by the Kolmogorov–Smirnov test. The data were expressed as arithmetic mean ± standard deviation (SD) and minimum/medium/maximum. The statistical difference among groups was compared using one-way ANOVA, followed by the post hoc Student–Newman–Keuls test or the Kruskal–Wallis test followed by the post hoc Mann–Whitney U test (where appropriate). Qualitative data between the control and treatment groups were analyzed by the Fisher test. The differences between the groups were considered statistically significant at *p* < 0.05.

## 3. Results

We have demonstrated that with BPC 157 therapy, after cauterization, the one remaining episcleral vein was upgraded to compensate otherwise inescapable venous congestion and counteract and avoid the increased intraocular pressure and consequent injurious course. The process of glaucoma development may be avoided (prophylactic application immediately before glaucoma surgery) and already advanced process markedly reversed (curative application 24 h after glaucoma surgery).

### 3.1. Intraocular Pressure, Pupillary Function, Fundoscopy and Histopathology of the Retina and Optic Nerve Head

#### 3.1.1. Intraocular Pressure

Cauterization of the three episcleral veins induced an immediate increase in intraocular pressure, which was sustained until the end of the experiment, at the end of the sixth post-injury week. On the other hand, all of the BPC 157 regimens (µg-, ng-, given locally, per-orally, intraperitoneally), started at first as prophylactic and later also as curative regimens, strongly reversed the increased intraocular pressure. The counteracting effect of BPC 157 application was immediate, already visible after the initial application, and was consistently maintained during the application period (Figure 1).

#### 3.1.2. Pupillary Function

Cauterization of the three episcleral veins that induced an immediate increase in intraocular pressure, as well as increased intraocular pressure until the end of the experiment at the end of the sixth post-injury week, simultaneously induced large mydriasis, particularly in the earlier post-injury period. In contrast, all of the BPC 157 regimens (µg-, ng-, given locally, per-orally, intraperitoneally), started as prophylactic and then also as curative regimens, strongly reversed the increased intraocular pressure and abrogated mydriasis. Same as the counteracting effect on the increased intraocular pressure, the counteracting effect of BPC 157 application on mydriasis was also immediate, already after the initial application, and was consistently maintained during the application period (Figure 2 and Figure 3).

#### 3.1.3. Fundoscopy and Histopathology of the Retina and Optic Nerve Head

##### Fundoscopy

Rapid increase in intraocular pressure, consistently increased intraocular pressure and simultaneous mydriasis are in accordance with fundoscopy findings. There were considerable changes in the posterior eye segment presentation, starting with the discrete generalized irregularity in the diameter of the retinal vessels with mild atrophy of the optic disc, less visible normal presentation of the choroidal blood vessels on the first post-injury day progressively leading to the strong generalized irregularity of the diameter of the blood vessels with severe atrophy of the optic disc and barely visible (extremely faint presentation) choroidal blood vessels (bright fundus background color) at the end of the sixth post-injury week. Contrarily, there was evidence that all BPC 157 regimens, both prophylactic and curative, strongly reversed the high intraocular pressure and abrogated mydriasis. Consequently, normal fundus presentation, normal retinal and choroidal blood vessel presentation and normal optic nerve presentation appeared in BPC 157-treated rats (Figure 4, Figure 5 and Figure 6).

##### Histopathology of the Retina

The cauterization of the three episcleral veins combined an immediate increase in intraocular pressure, increased intraocular pressure until the end of the experiment at the end of the sixth post-injury week, mydriasis, and finally, established by fundoscopy, strong generalized irregularity of the diameter of the blood vessels with severe atrophy of the optic disc and barely visible (extremely faint presentation) choroidal blood vessels (bright fundus background color) at the end of the sixth post-injury week. Thus, the degeneration of ganglion cells (i.e., degeneration in retinal ganglion cell layer, inner and outer nuclear layer) and whole retina appeared as the final downhill outcome. Contrarily, the previously described beneficial counteracting effect of all BPC 157 regimens (µg-, ng-, given locally, per-orally, intraperitoneally), both prophylactic and later, curative, resulted in preserved ganglion cells presentation (Figure 7, Figure 8, Figure 9, Figure 10 and Figure 11).

##### Histopathology of the Optic Nerve Head

As can be noted at the end of the experiments, at the end of the sixth post-operative week the conclusive point of the cauterization of the three episcleral veins was the induced progressive optic nerve head excavation and reduction in optic nerve thickness. Contrarily, the previously described beneficial counteracting effect of all BPC 157 regimens (µg-, ng-, given locally, per-orally, intraperitoneally), both prophylactic and later, curative, caused BPC 157 rats to have no optic nerve head excavation whatsoever, and optic nerve thickness was preserved (Figure 12, Figure 13 and Figure 14).

In summary, after the cauterization of three episcleral veins, intraocular pressure in all BPC 157 regimens was immediately normalized. BPC 157-treated rats exhibited normal pupil diameter, preserved ganglion cells, inner nuclear layer and outer nuclear layer, and optic nerve presentation, normal fundus presentation, normal retinal and choroidal blood vessel presentation and normal optic nerve presentation. As leading symptoms, increased intraocular pressure and mydriasis, as well as degeneration of retinal ganglion cells, optic nerve head excavation and reduction in optic nerve thickness, generalized severe irregularity of retinal vessels, faint presentation of choroidal vessels and severe optic nerve disc atrophy, were all counteracted.

## 4. Discussion

BPC 157 therapy counteracted the increased intraocular pressure induced by three episcleral vein cauterization in rats. This occurs with BPC 157 therapy in equal effect when given prophylactically (immediately before cauterization) or therapeutically (i.e., 24 h after cauterization, upon the established increased intraocular pressure), applied locally, intraperitoneally and per-orally and in suitable dose (µg-, ng-) range. As in the other vessel occlusion models [7,8,9,10,11,12,13], this occurs in seconds. That is too fast for aqueous dynamics [52] to play a role in the BPC 157-induced decrease in the intraocular pressure and the normalization of intraocular pressure in the glaucomatous rats. It may be that the recruitment of the remaining vessel(s) as alternative upgraded bypassing pathway drains out the residual blood volume of the eye, resulting in a rapid net decrease in blood volume and an equally rapid fall in the increased intraocular pressure [52]. Previously, ascribed to the resolved Virchow in BPC 157 therapy, these meant the prevented/counteracted whole occlusion syndrome and portal, caval and intracranial hypertension in particular (i.e., bile duct ligation-cirrhosis, vessel occlusion syndromes (Pringle maneuver, Budd–Chiari, superior mesenteric artery and/or vein, superior sagittal sinus)) [7,8,9,10,11,12,13]. Together, they may provide indicative arguments, as in other vessel occlusion studies [36,37,38,39,40], leading to the prevented/counteracted elevated intraocular pressure in glaucomatous rats. These are practical points that support each other. The follow-up is the consistently abrogated consequent glaucoma-like injurious course. Thereby, a recovered circulation provides a continuous flow of aqueous humor for nourishing the avascular cornea, lens and vitreous compartment, the production of aqueous humor (and its ease of egress from the eye), the maintenance of adequate intraocular pressure and avoidance of a compressing force on the ocular blood vessels and optic nerve [52]. Consequent are normal pupil presentation, preserved ganglion cells and optic nerve presentation, normal fundus presentation, normal retinal and choroidal blood vessel presentation and normal optic nerve presentation in BPC 157-treated rats. This effect reversed the general poor presentation of the rats with three cauterized episcleral veins. Without therapy, there were the constantly increased intraocular pressure and mydriasis, degeneration of ganglion cells (i.e., degeneration in inner and outer nuclear layer), optic nerve head excavation and reduction in optic nerve thickness, generalized severe irregularity of the retinal vessels, faint presentation of choroidal vessels and severe optic nerve disc atrophy. As mentioned, these disturbances were all counteracted, as demonstrated.

In support, as in the glaucoma-like damage induced by the three episcleral veins cauterization, BPC 157 also counteracted L-NAME–induced rat retinal ischemia [17]. Similarly to the present therapy findings in rats with cauterized episcleral veins, the L-NAME–induced rat retinal ischemia [17] fundoscopy demonstrated, quite rapidly, normal eye background and choroidal blood vessels. There were normal eye background and choroidal blood vessels maintained in all subsequent periods. In particular, the damage of inner plexiform layer and inner nuclear layer was counteracted and normal retinal thickness revealed, at 1, 2 and 4 weeks, and otherwise poor behavioral presentation rescued (i.e., the highly expressed freezing behavior was completely reversed after BPC 157 therapy was given) [17].

Also, there are previous supporting BPC 157 pupillary effects and the counteraction of atropine-induced mydriasis [44]. Thereby, the recovering normal intraocular pressure in glaucomatous rats with episcleral vein cautery by BPC 157 means the eliminated blocking drainage of the intraocular fluid from the angle of the anterior chamber in dilated iris goes together with the counteraction of the raised intraocular pressure caused by episcleral vein cautery. Together, these findings are likely indicative of the open-angle close glaucoma therapy and acute angle-closure glaucoma, maintaining pupillary function by BPC 157 [44]. Further arguments are the mydriasis consequent to elevation of the intraocular pressure also in humans [53,54]. Such findings suggest a combined consistent effect of BPC 157 application (note, BPC 157 does not affect normal intraocular pressure [17], and it does not affect normal pupil diameter [44]) may be advantageous in providing both the ambiguous pressure-lowering effect and mydriasis counteracting effect of the standard anti-glaucoma agents [55,56]. On the increased intraocular pressure, there were delayed onset [56], effect absent [57,58] or even opposite effect [59]. On the presented mydriasis, there were miosis delayed onset [60] and rebounded mydriasis [61].

In addition, the essential upgrading of the remained episcleral vein with BPC 157 therapy would consider also the possible contribution of the unconventional uveoscleral pathway (initial relaxation of the ciliary muscle and subsequently decreased resistance in the uveoscleral pathways [58]). There is supporting evidence that the ophthalmic vein can also be pointed out as rescuing pathway in rats with central vein (superior sagittal sinus) occlusion [13]. The bypassing loop was along angular vein, facial anterior and posterior vein, facial vein, through external jugular vein, subclavian vein through superior vena cava, which led to rapid attenuation of the brain swelling and rapid elimination of the increased pressure in the ligated superior sagittal sinus [13]. Simultaneously, in the periphery, there were activation of azygos vein, rapid elimination of the severe portal and caval hypertension and aortal hypotension, and rapid collateral vessel recruitment. Abrogated venous and arterial thrombosis and recovery of the organ lesions occurred consistently both peripherally and centrally [13]. Furthermore, there is evidence that BPC 157 counteracted stroke, given in reperfusion, after clamping of the common carotid arteries (i.e., both early and delayed neural hippocampal damage and achieving full functional recovery) and ameliorated other central [62,63,64,65,66,67,68,69,70] and peripheral [71] neuronal damage, in particularly those in rats with peripheral or central major vessel occlusion-induced syndromes [7,8,9,10,11,12,13]. This may suggest that, in this way, the neuroprotective effect of BPC 157 administration would also counteract retinal disturbances in glaucomatous disturbances, even with normal intraocular pressure, since BPC 157 might counteract retrobulbar L-NAME application-induced progressive severe atrophy of optic nerve in rats with the normal intraocular pressure [17]. Thus, the decrease in elevated intraocular pressure and attenuation of the subsequent injurious course in rats with cauterized episcleral veins by BPC 157 proves its applicable background.

Summarizing, the perilous analogous relation in rats with three cauterized episcleral veins clearly established [45] that, as there were more lesions (i.e., more cauterized episcleral veins), the intraocular pressure increased further [45]. This maxim would verify the highest compensatory capacity of the therapy. Thereby, in the BPC 157-treated rats with three cauterized episcleral veins, there were the normal intraocular pressure, the promptly and sustainably upgraded left episcleral vein as the cause–consequence of the normal intraocular pressure, normal pupil presentation, preserved ganglion cells and optic nerve presentation, normal fundus presentation, normal retinal and choroidal blood vessel presentation and normal optic nerve presentation. A particular point is revealing and applies this concept in practice [72], also in various species [73], since it has a very safe profile (LD1 could be not achieved) [72], a point recently confirmed in a large study by Xu et al. [74]. There are consistently effective used ranges of BPC 157 (µg-ng) application and used regimens, which may support each other’s effects [3,4,5,6]. On the other hand, while establishing the general therapy principle, this observational study did not investigate the particular molecular mechanism. However, considering the recently postulated role of gap junction in cell death and neuromodulation in the retina [75], we should emphasize the very recent demonstration of the mitigated leaky gut syndrome, which revealed BPC 157 activity as a stabilizer of cellular junction, via increasing tight junction protein ZO-1 expression and transepithelial resistance [22]. There were inhibition of mRNA of inflammatory mediators (iNOS, IL-6, IFNγ and TNF-α), increased expression of HSP 70 and 90, and antioxidant proteins, such as HO-1, NQO-1, glutathione reductase, glutathione peroxidase 2 and GST-pi [22]. It is of note that the antioxidant effect of BPC 157 [32,33,34] occurs in both ischemic and reperfusion conditions in the various tissues (i.e., colon, duodenum, cecum, liver and veins) and plasma [7,8,9,10,11,12,36,37,38,39,40]. Likewise, in the vessel that provides an alternative operating pathway (i.e., left ovarian vein as the key for resolving the infrarenal occlusion-induced inferior vena cava syndrome in rats) [7], BPC 157 exhibited a specific effect on the *Egr*, *Nos*, *Srf*, *Vegfr*, *Akt1*, *Plcɣ* and *Kras* pathways, serving as a possible highlight of how BPC 157 may act.

Thus, we report BPC 157 and rat glaucoma induced by episcleral veins cauterization as major vessel occlusion syndrome that was resolved in this study, with particular respect to counteraction of all consequences (i.e., increased intraocular pressure, mydriasis, retinal and optic nerve lesions) and therapy of BPC 157.

## Figures and Tables

**Figure 1 biomedicines-10-00089-f001:**
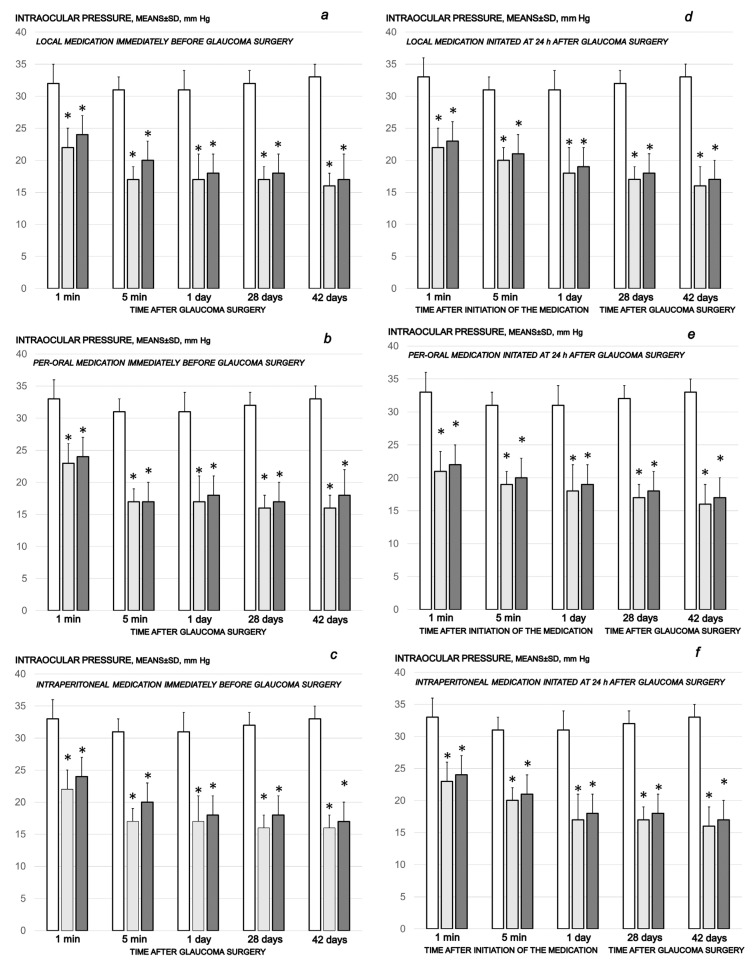
Cauterization of the three episcleral veins induced increased intraocular pressure, which remained sustained until the end of the of the experiment at the end of the sixth post-injury week. Counteracting effect of BPC 157 medication daily regimen, prophylactic (**a**–**c**) or curative (**d**–**f**) (gray bars, light gray bars µg-regimen, dark gray bars ng-regimen (0.4 µg/eye, 0.4 ng/eye; 10 µg/kg, 10 ng/kg) given locally as eye drops in each eye (**a**,**d**), intraperitoneally (last application at 24 h before sacrifice) (**b**,**e**) or per-orally in drinking water (0.16 µg/mL, 0.16 ng/mL, 12 mL/rat until the sacrifice, first application given intragastrically) (**c**,**f**), started prophylactically (immediately before glaucoma surgery) or as curative treatment (at 24 h after glaucoma surgery). Controls (white bars) simultaneously received an equal volume of distilled water (two drops/eye), saline intraperitoneal application (5 mL/kg) or per-oral drinking water (12 mL/day/rat). A total of 6 rats/group/interval. * *p* < 0.05, at least, vs. control.

**Figure 2 biomedicines-10-00089-f002:**
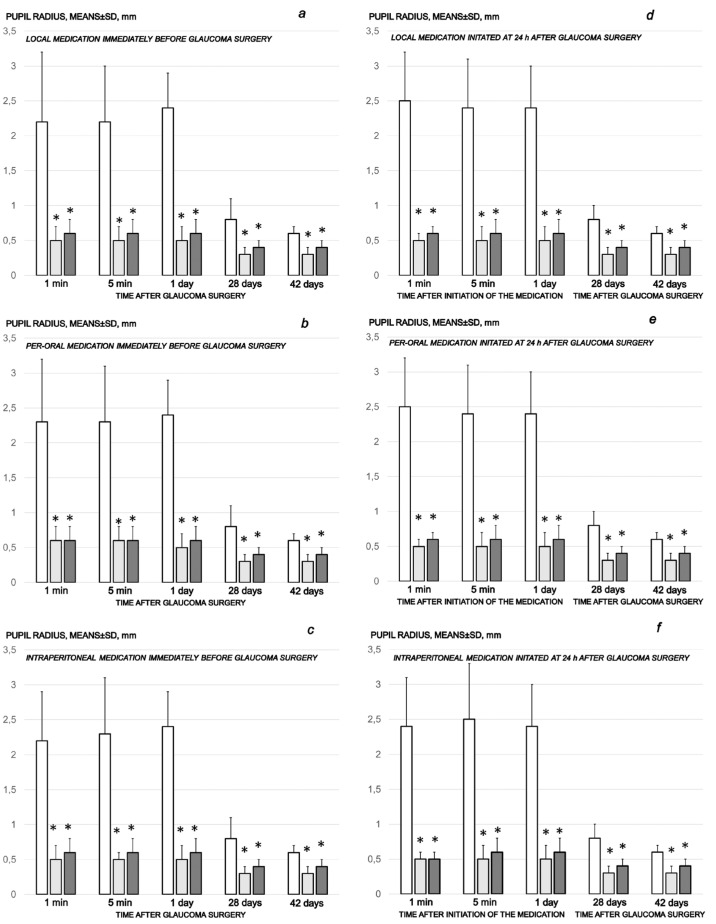
Cauterization of the three episcleral veins induced mydriasis, enormous at the earlier post-injury periods, and present until the end of the experiment at the end of the sixth post-injury week. Counteracting effect of BPC 157 medication daily regimen, prophylactic (**a**–**c**) or curative (**d**–**f**) (gray bars, light gray bars µg-regimen, dark gray bars ng-regimen (0.4 µg/eye, 0.4 ng/eye; 10 µg/kg, 10 ng/kg) given locally as eye drops in each eye (**a**,**d**), intraperitoneally (last application at 24 h before sacrifice) (**b**,**e**) or per-orally in drinking water (0.16 µg/mL, 0.16 ng/mL, 12 mL/rat until the sacrifice, first application given intragastrically) (**c**,**f**)), started prophylactically (immediately before glaucoma surgery) or as curative treatment (at 24 h after glaucoma surgery). Controls (white bars) simultaneously received an equal volume of distilled water (two drops/eye), saline intraperitoneal application (5 mL/kg) or per-oral drinking water (12 mL/day/rat). A total of 6 rats/group/interval. * *p* < 0.05, at least, vs. control.

**Figure 3 biomedicines-10-00089-f003:**
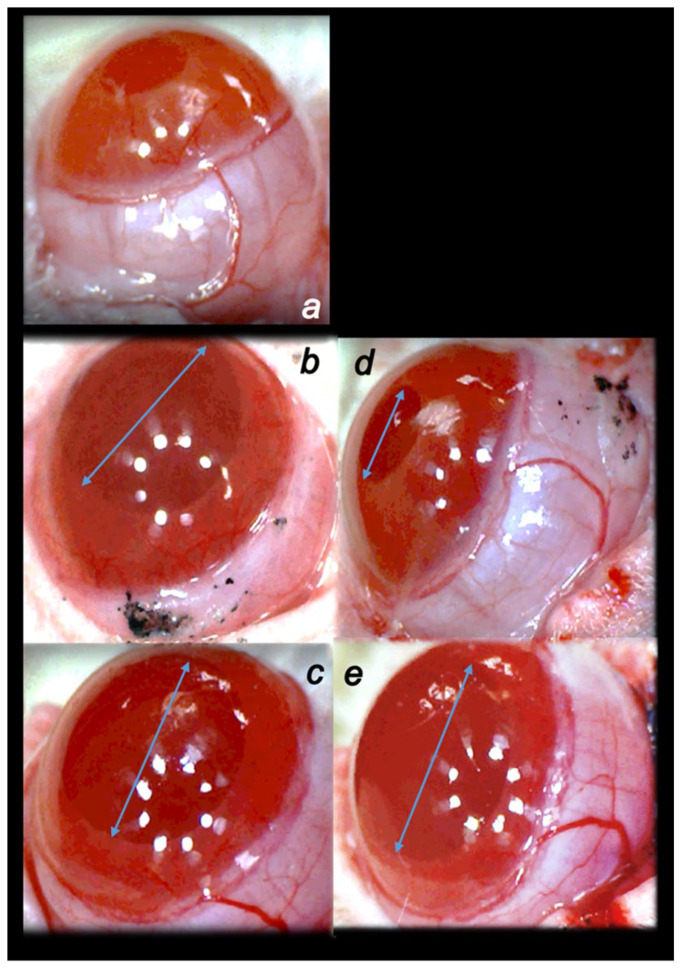
Illustrative presentation of the pupil changes. (**a**). Normal pupil presentation in healthy rats ((**a**), white letter). (**b**). Persisting mydriasis (i.e., at 24 h post-injury) as effect of cauterization of the three episcleral veins. (**c**). Persisting mydriasis (i.e., at 24 h post-injury) as effect of cauterization of the three episcleral veins. (**d**). Counteraction of the previously persisting mydriasis (i.e., throughout 24 h post-injury) as effect of cauterization of the three episcleral veins upon BPC 157 therapy. (**e**). Upon distilled water eye therapy, unchanged presentation of continuously persisting mydriasis (i.e., throughout 24 h post-injury) as effect of cauterization of the three episcleral veins. Regular effect of cauterization of the three episcleral veins ((**b**,**c**)*,* black letters), persisting mydriasis, illustration at 24 h post-injury (arrows), and effect of therapy application, BPC 157 (mydriasis counteraction) ((**d**), black letter) ((**b**)→(**d**)) or distilled water eye drops ((**c**), black letter) (persisted mydriasis) ((**c**)→(**e**)) (at 1 min after therapy). A similar effect was noted with all BPC 157 regimens. USB microscope camera Veho Discovery VMS-004 Deluxe. A total of 6 rats/group/interval.

**Figure 4 biomedicines-10-00089-f004:**
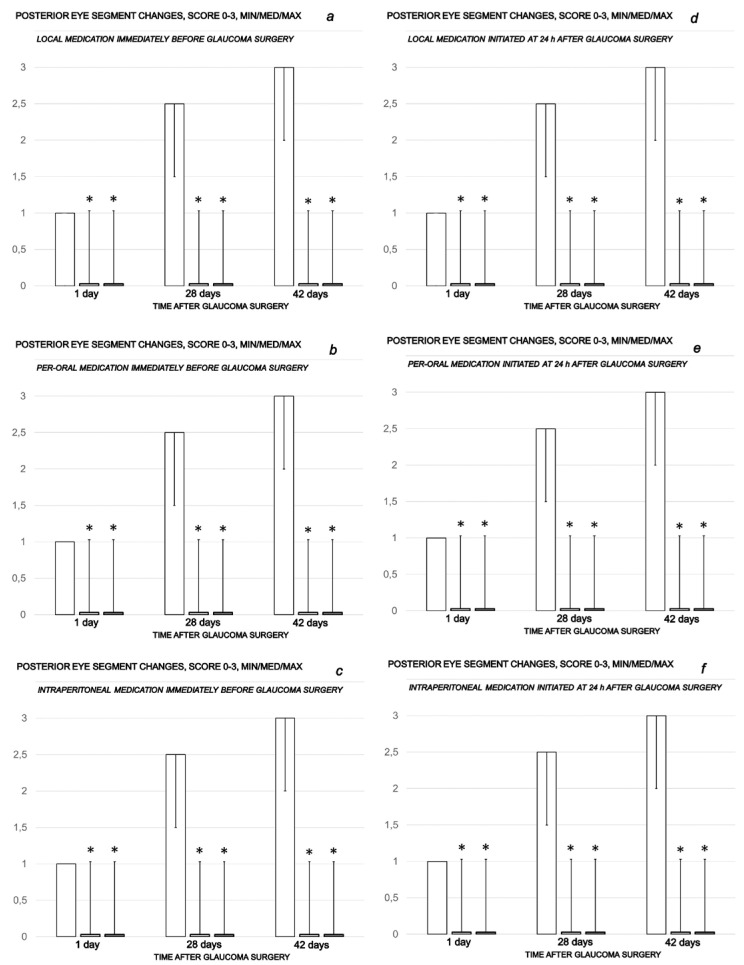
Cauterization of the three episcleral veins induced posterior eye segment changes assessed by fundoscopy, scored 0–3, min/med/max. Counteracting effect of BPC 157 medication daily regimen, prophylactic (**a**–**c**) or curative (**d**–**f**) (gray bars, light gray bars µg-regimen, dark gray bars ng-regimen (0.4 µg/eye, 0.4 ng/eye; 10 µg/kg, 10 ng/kg) given locally as eye drops in each eye (**a**,**d**), intraperitoneally (last application at 24 h before sacrifice) (**b**,**e**) or per-orally in drinking water (0.16 µg/mL, 0.16 ng/mL, 12 mL/rat until the sacrifice, first application given intragastrically) (**c**,**f**)), started prophylactically (immediately before glaucoma surgery) or as curative treatment (at 24 h after glaucoma surgery, assessment at 1 h after application). Controls (white bars) simultaneously received an equal volume of distilled water (two drops/eye), saline intraperitoneal application (5 mL/kg) or per-oral drinking water (12 mL/day/rat). A total of 6 rats/group/interval. * *p* < 0.05, at least, vs. control.

**Figure 5 biomedicines-10-00089-f005:**
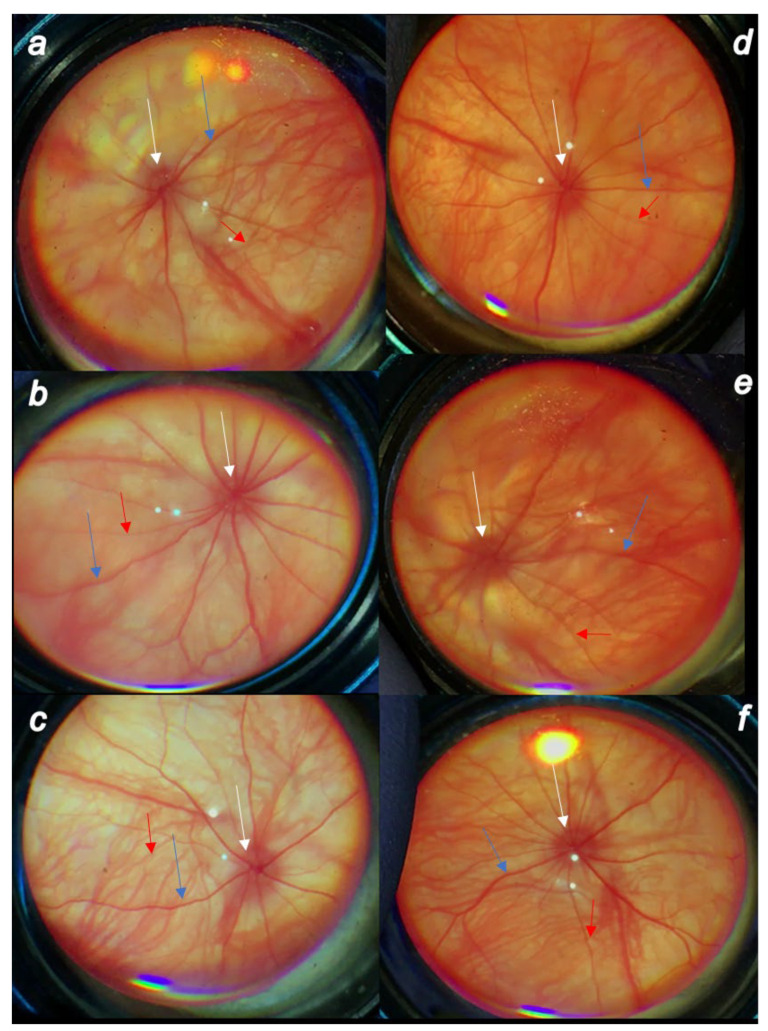
Cauterization of the three episcleral veins induced posterior eye segment changes (veins (blue arrows), arteries (red arrows), optic disc (white arrows)) assessed by fundoscopy. Prophylactic regimen. (**a**). First post-injury day, regular course. (**b**). Fourth post-operative week, regular course. (**c**). Sixth post-injury week, regular course. (**d**). First post-injury day, course in BPC 157-treated rats. (**b**). Fourth post-operative week in BPC 157-treated rats. (**c**). Sixth post-injury week in BPC 157-treated rats. There were discrete generalized irregularity in the diameter of the retinal vessels with mild atrophy of the optic disc, less visible normal presentation of the choroidal blood vessels at first post-injury day (**a**), progressively leading to evident worsening at the fourth post-operative week (**b**) with the strong generalized irregularity of the diameter of the blood vessels with severe atrophy of the optic disc and barely visible (extremely faint presentation) choroidal blood vessels (bright fundus background color) at the end of the sixth post-injury week (**c**). Contrarily, consequent to the evidence that all BPC 157 regimens, both prophylactic regimen and later, curative regimen, strongly reversed the increased intraocular pressure and abrogated mydriasis, illustrated are the normal fundus presentation, the presentation of the normal retinal and choroidal blood vessel presentation, and normal optic nerve presentation at the first post-operative day (**d**), fourth post-operative week (**e**) and sixth post-operative week (**f**) in the rats that started with BPC 157 therapy immediately before glaucoma surgery. A similar effect was noted with all BPC 157 regimens. A total of 6 rats/group/interval.

**Figure 6 biomedicines-10-00089-f006:**
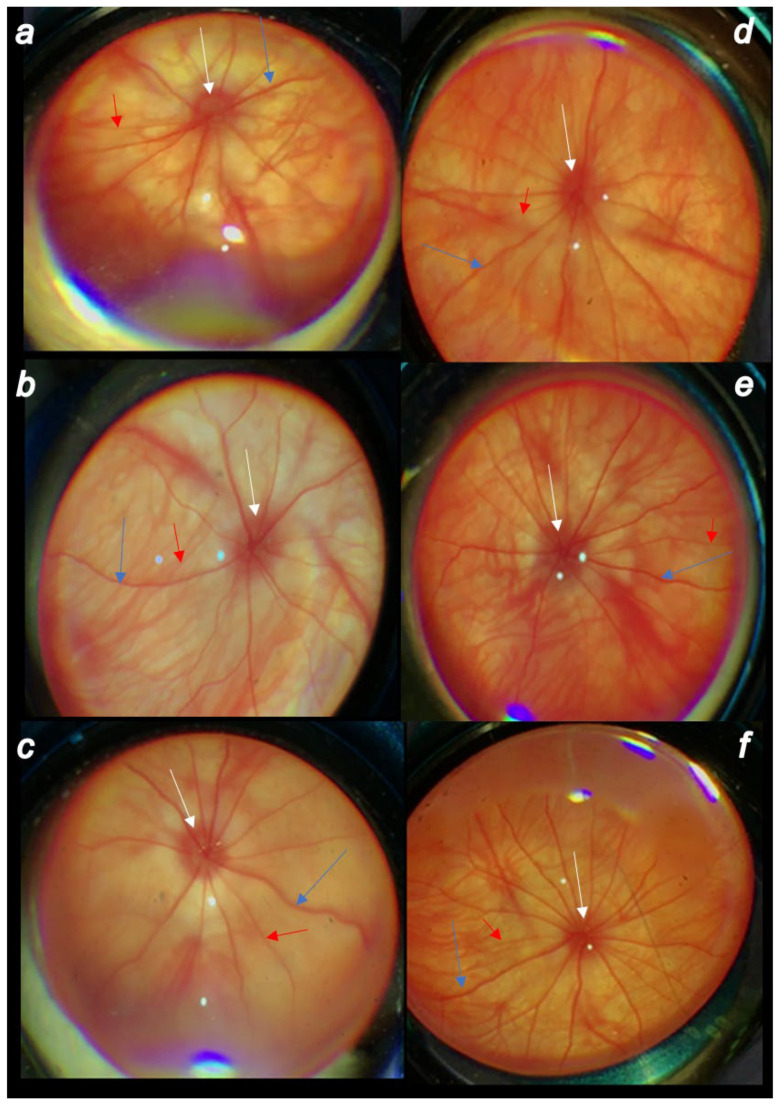
Cauterization of the three episcleral veins induced posterior eye segment changes (veins (blue arrows), arteries (red arrows), optic disc (white arrows)) assessed by fundoscopy. Therapeutic regimen. (**a**). First post-injury day, regular course. (**b**). Fourth post-operative week, regular course. (**c**). Sixth post-injury week, regular course. (**d**). First post-injury day, course in BPC 157-treated rats. (**b**). Fourth post-operative week in BPC 157-treated rats. (**c**). Sixth post-injury week in BPC 157-treated rats. There were the discrete generalized irregularity in the diameter of the retinal vessels with mild atrophy of the optic disc, less visible normal presentation of the choroidal blood vessels at first post-injury day (**a**), progressively leading to evident worsening at the fourth post-operative week (**b**) with the strong generalized irregularity of the diameter of the blood vessels with severe atrophy of the optic disc and barely visible (extremely faint presentation) choroidal blood vessels (bright fundus background color) at the end of the sixth post-injury week (**c**). Contrarily, consequent to the evidence that all BPC 157 regimens, both prophylactic regimen and later, curative regimen, strongly reversed the increased intraocular pressure and abrogated mydriasis, illustrated are the normal fundus presentation, the presentation of the normal retinal and choroidal blood vessel presentation, and normal optic nerve presentation at the first post-operative day (**d**) (1 h after BPC 157 therapy application), fourth post-operative week (**e**) and sixth post-operative week (**f**) in the rats that started with BPC 157 therapy at post-surgery day 1. A similar effect was noted with all BPC 157 regimens. A total of 6 rats/group/interval.

**Figure 7 biomedicines-10-00089-f007:**
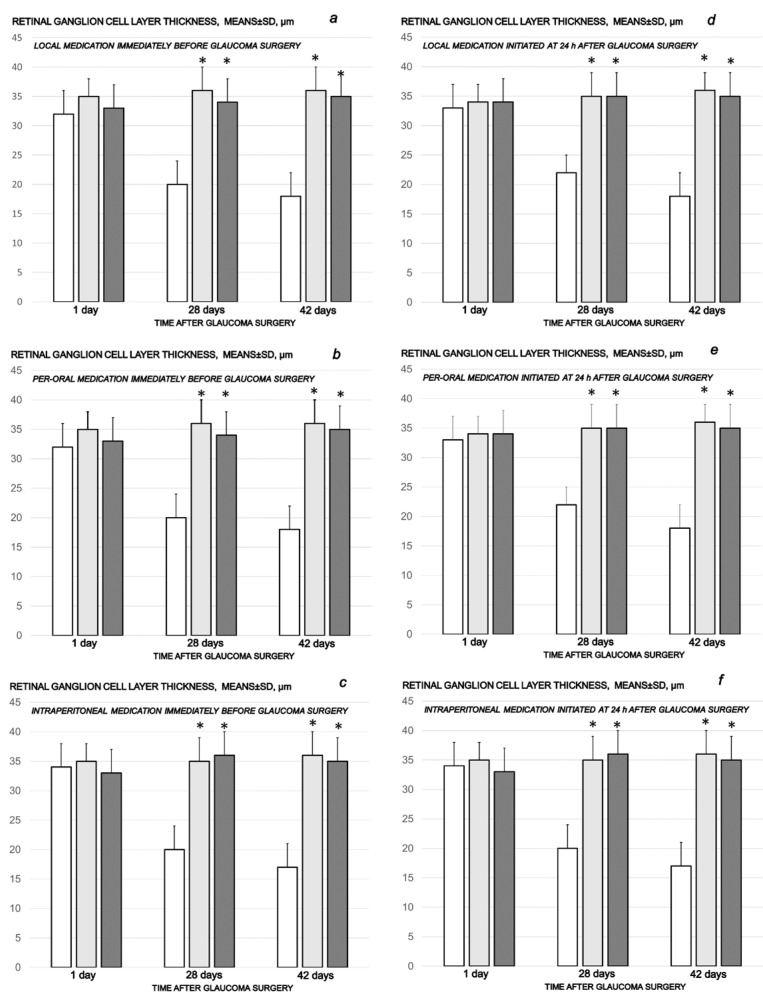
Cauterization of the three episcleral veins induced a decrease in retinal ganglion cell layer thickness, means ± SD, µn, at the first post-operative day and at the end of the fourth and sixth post-injury week. Counteracting effect of BPC 157 medication daily regimen, prophylactic (**a**–**c**) or curative (**d**–**f**) (gray bars, light gray bars µg-regimen, dark gray bars ng-regimen (0.4 µg/eye, 0.4 ng/eye; 10 µg/kg, 10 ng/kg) given locally as eye drops in each eye (**a**,**d**), intraperitoneally (last application at 24 h before sacrifice) (**b**,**e**) or per-orally in drinking water (0.16 µg/mL, 0.16 ng/mL, 12 mL/rat until the sacrifice, first application given intragastrically) (**c**,**f**)), started prophylactically (immediately before glaucoma surgery) or as curative treatment (at 24 h after glaucoma surgery). Controls (white bars) simultaneously received an equal volume of distilled water (two drops/eye), saline intraperitoneal application (5 mL/kg) or per-oral drinking water (12 mL/day/rat). A total of 6 rats/group/interval. * *p* < 0.05, at least, vs. control.

**Figure 8 biomedicines-10-00089-f008:**
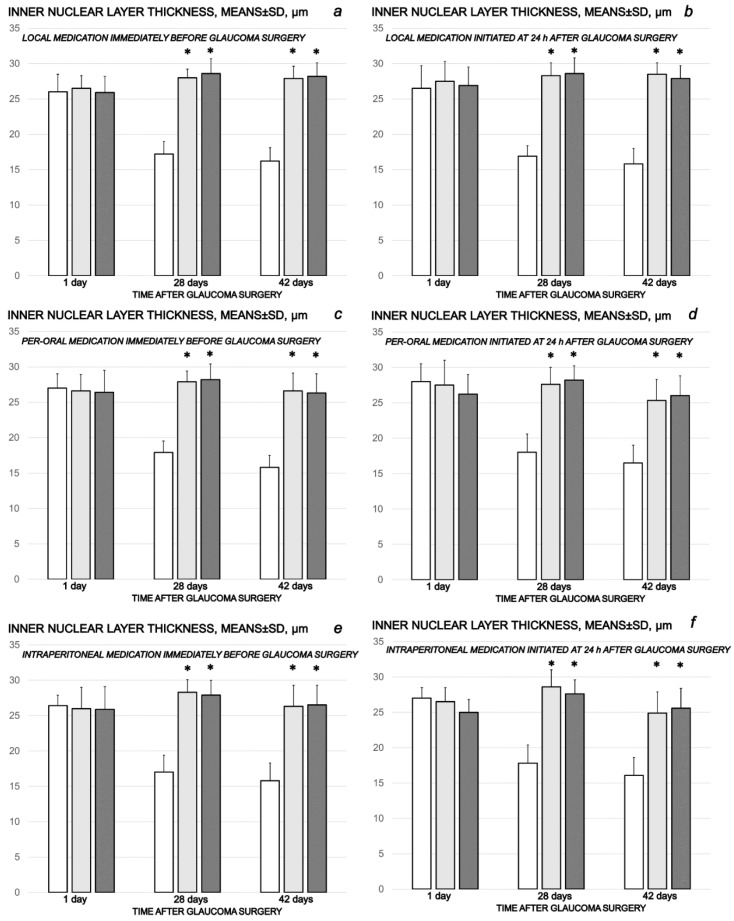
Cauterization of the three episcleral veins induced a decrease in retinal inner nuclear layer thickness, means ± SD, µn, at the first post-operative day and at the end of the fourth and sixth post-injury week. Counteracting effect of BPC 157 medication daily regimen, prophylactic (**a**–**c**) or curative (**d**–**f**) (gray bars, light gray bars µg-regimen, dark gray bars ng-regimen (0.4 µg/eye, 0.4 ng/eye; 10 µg/kg, 10 ng/kg) given locally as eye drops in each eye (**a**,**d**), intraperitoneally (last application at 24 h before sacrifice) (**b**,**e**) or per-orally in drinking water (0.16 µg/mL, 0.16 ng/mL, 12 mL/rat until the sacrifice, first application given intragastrically) (**c**,**f**)), started prophylactically (immediately before glaucoma surgery) or as curative treatment (at 24 h after glaucoma surgery). Controls (white bars) simultaneously received an equal volume of distilled water (two drops/eye), saline intraperitoneal application (5 mL/kg) or per-oral drinking water (12 mL/day/rat). A total of 6 rats/group/interval. * *p* < 0.05, at least, vs. control.

**Figure 9 biomedicines-10-00089-f009:**
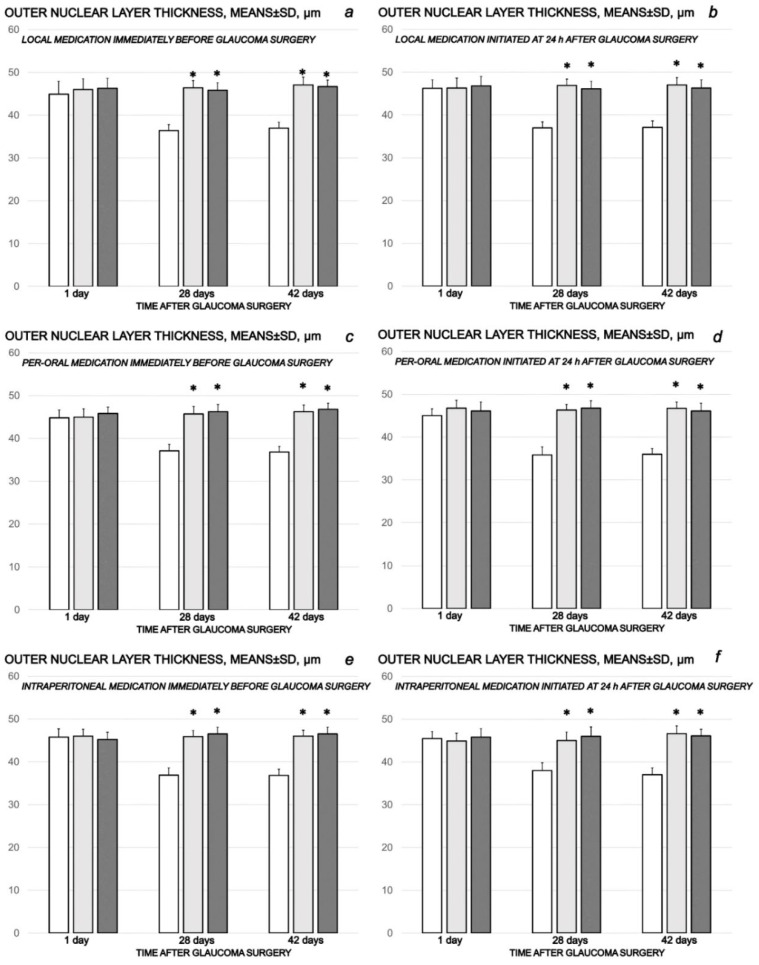
Cauterization of the three episcleral veins induced a decrease in retinal outer nuclear layer thickness, means ± SD, µn, at the first post-operative day and at the end of the fourth and sixth post-injury week. Counteracting effect of BPC 157 medication daily regimen, prophylactic (**a**–**c**) or curative (**d**–**f**) (gray bars, light gray bars µg-regimen, dark gray bars ng-regimen (0.4 µg/eye, 0.4 ng/eye; 10 µg/kg, 10 ng/kg) given locally as eye drops in each eye (**a**,**d**), intraperitoneally (last application at 24 h before sacrifice) (**b**,**e**) or per-orally in drinking water (0.16 µg/mL, 0.16 ng/mL, 12 mL/rat until the sacrifice, first application given intragastrically) (**c**,**f**)), started prophylactically (immediately before glaucoma surgery) or as curative treatment (at 24 h after glaucoma surgery). Controls (white bars) simultaneously received an equal volume of distilled water (two drops/eye), saline intraperitoneal application (5 mL/kg) or per-oral drinking water (12 mL/day/rat. A total of 6 rats/group/interval. * *p* < 0.05, at least, vs. control.

**Figure 10 biomedicines-10-00089-f010:**
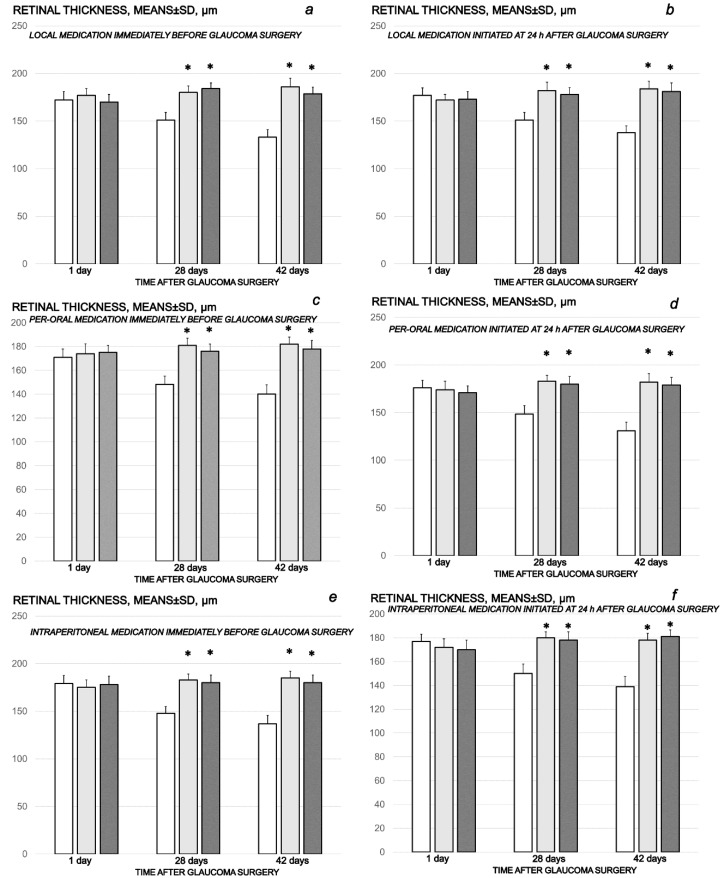
Cauterization of the three episcleral veins induced a decrease in retinal thickness, means ± SD, µn, at the first post-operative day and at the end of the fourth and sixth post-injury week. Counteracting effect of BPC 157 medication daily regimen, prophylactic (**a**–**c**) or curative (**d**–**f**) (gray bars, light gray bars µg-regimen, dark gray bars ng-regimen (0.4 µg/eye, 0.4 ng/eye; 10 µg/kg, 10 ng/kg) given locally as eye drops in each eye (**a**,**d**), intraperitoneally (last application at 24 h before sacrifice) (**b**,**e**) or per-orally in drinking water (0.16 µg/mL, 0.16 ng/mL, 12 mL/rat until the sacrifice, first application given intragastrically) (**c**,**f**)), started prophylactically (immediately before glaucoma surgery) or as curative treatment (at 24 h after glaucoma surgery). Controls (white bars) simultaneously received an equal volume of distilled water (two drops/eye), saline intraperitoneal application (5 mL/kg) or per-oral drinking water (12 mL/day/rat. A total of 6 rats/group/interval. * *p* < 0.05, at least, vs. control.

**Figure 11 biomedicines-10-00089-f011:**
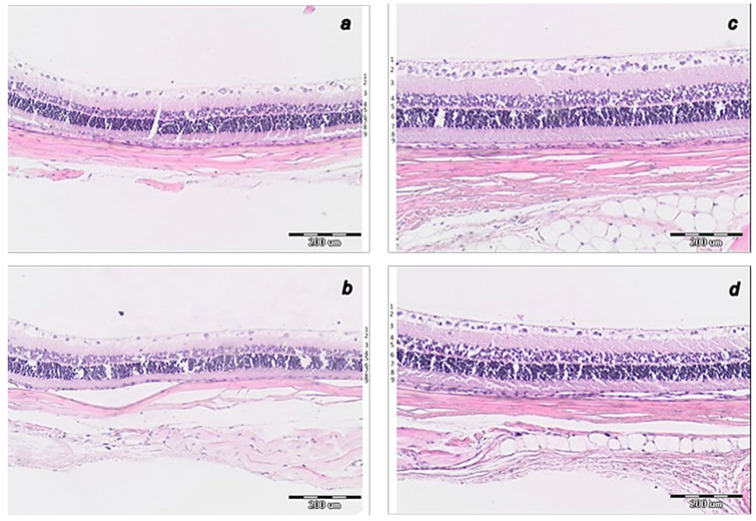
Histology of central part of transverse section of the retina 0.4–0.7 mm on the temporal side of the optic disc. (**a**). Rat retina after episcleral vein cauterization at week 4 in controls. (**b**). Rat retina after episcleral vein cauterization at week 6 in controls. (**c**). Rat retina after episcleral vein cauterization at week 4 in BPC 157-treated rats. (**d**). Rat retina after episcleral vein cauterization at week 6 in BPC 157-treated rats. (HE staining, magnification ×20, scale bar 200 μm). Transverse section of the retina (0.4–0.7 mm on the temporal side of the optic disc) showing a strict difference in the retinal layers and full retina thickness in the rats that received saline and those that received BPC 157. More regular inner and outer nuclear layer and more regular distribution of ganglion cells, preserved thickness of retina, the inner and outer plexiform layer at week 4 in BPC 157-treated rats (**c**) than in control rats (**a**). At week 6, BPC 157-treated rats show the preserved thickness of the whole retina and also the inner plexiform layer and inner nuclear layer and its organization (**d**). Contrarily, degeneration of ganglion cells in control group is the most evident also as degeneration in inner and outer nuclear layer (**b**). Retinal layers as follows: 1—internal limiting membrane; 2—nerve fiber and ganglion cell layers; 3—inner plexiform layer; 4—inner nuclear layer; 5—outer plexiform layer; 6—outer nuclear layer; 7—outer limiting membrane; 8—photoreceptor layer; 9—pigment epithelium. A similar effect was noted with all BPC 157 regimens. A total of 6 rats/group/interval.

**Figure 12 biomedicines-10-00089-f012:**
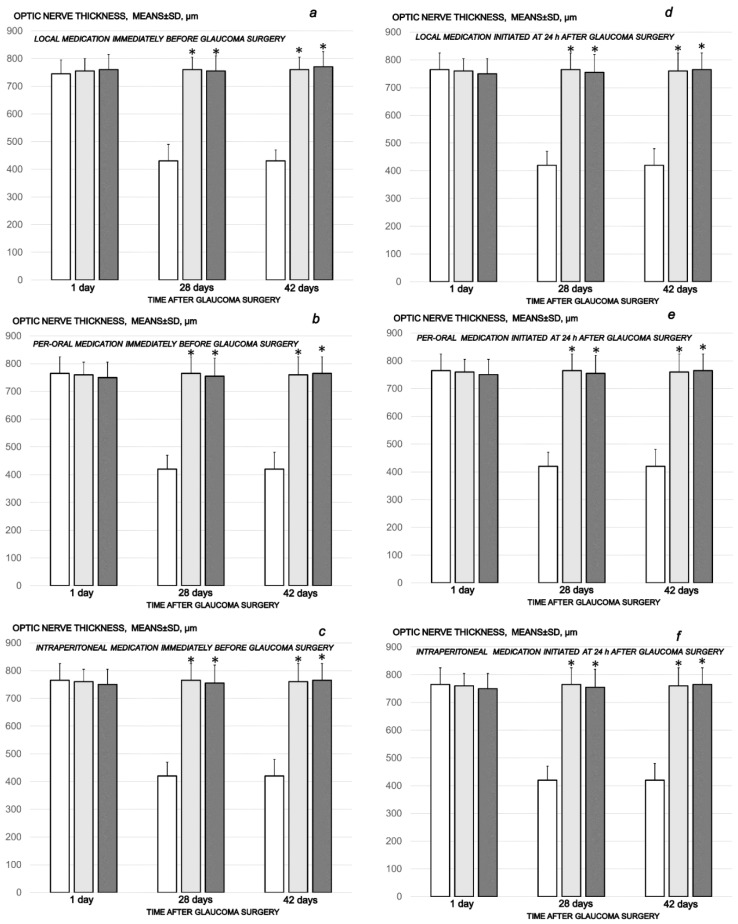
Cauterization of the three episcleral veins induced a decrease in optic nerve thickness, means ± SD, µm, at the first post-operative day and at the end of the fourth and sixth post-injury week. Counteracting effect of BPC 157 medication daily regimen, prophylactic (**a**–**c**) or curative (**d**–**f**) (gray bars, light gray bars µg-regimen, dark gray bars ng-regimen (0.4 µg/eye, 0.4 ng/eye; 10 µg/kg, 10 ng/kg) given locally as eye drops in each eye (**a**,**d**), intraperitoneally (last application at 24 h before sacrifice) (**b**,**e**) or per-orally in drinking water (0.16 µg/mL, 0.16 ng/mL, 12 mL/rat until the sacrifice, first application given intragastrically) (**c**,**f**)), started prophylactically (immediately before glaucoma surgery) or as curative treatment (at 24 h after glaucoma surgery). Controls (white bars) simultaneously received an equal volume of distilled water (two drops/eye), saline intraperitoneal application (5 mL/kg) or per-oral drinking water (12 mL/day/rat). * minimum *p* < 0.05 vs. control. A total of 6 rats/group/interval.

**Figure 13 biomedicines-10-00089-f013:**
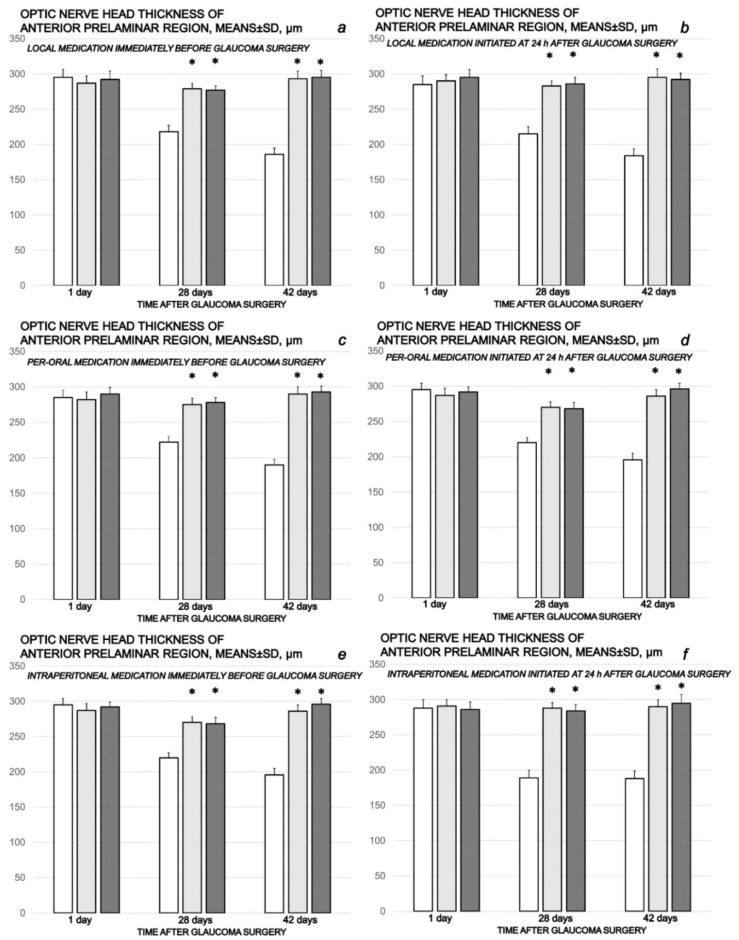
Cauterization of the three episcleral veins induced a decrease in optic nerve head thickness of anterior prelaminar region, means ± SD, µm, at the first post-operative day and at the end of the fourth and sixth post-injury week. Counteracting effect of BPC 157 medication daily regimen, prophylactic (**a**–**c**) or curative (**d**–**f**) (gray bars, light gray bars µg-regimen, dark gray bars ng-regimen (0.4 µg/eye, 0.4 ng/eye; 10 µg/kg, 10 ng/kg) given locally as eye drops in each eye (**a**,**d**), intraperitoneally (last application at 24 h before sacrifice) (**b**,**e**) or per-orally in drinking water (0.16 µg/mL, 0.16 ng/mL, 12 mL/rat until the sacrifice, first application given intragastrically) (**c**,**f**)), started prophylactically (immediately before glaucoma surgery) or as curative treatment (at 24 h after glaucoma surgery). Controls (white bars) simultaneously received an equal volume of distilled water (two drops/eye), saline intraperitoneal application (5 mL/kg) or per-oral drinking water (12 mL/day/rat). * minimum *p* < 0.05 vs. control. A total of 6 rats/group/interval.

**Figure 14 biomedicines-10-00089-f014:**
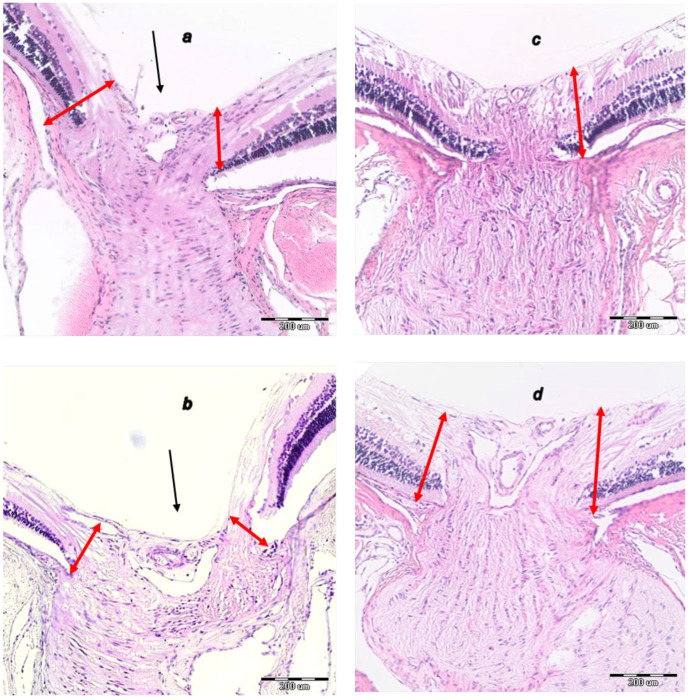
Histopathology of the optic nerve head. (**a**). Optic nerve head of rat retina after episcleral vein cauterization at week 4 in controls. (**b**). Optic nerve head of rat retina after episcleral vein cauterization at week 4 in controls. (**c**). Optic nerve head of rat retina after episcleral vein cauterization at week 4 in BPC 157-treated rats. (**d**). Optic nerve head of rat retina after episcleral vein cauterization at week 6 in BPC 157-treated rats. Optic nerve head excavation (black arrow) and thickness of anterior prelaminar region (double red arrow); HE staining, magnification ×20, scale bar 200 μm). Transverse section in the center of the optic nerve head showing a strict difference in the optic nerve head and optic nerve in the rats that received saline and those that received BPC 157. Optic nerve head in rats that received BPC 157 at week 4 (**c**) presented as not excavated, while optic nerve thickness and optic nerve head of anterior prelaminar region were preserved. On the contrary, in rats that received saline, at week 4 (**a**) optic nerve head excavation was present, while optic nerve and optic nerve head of anterior prelaminar region thickness was reduced. In rats that received BPC 157 at week 6 (**d**), optic nerve head excavation was not excavated at all, and optic nerve thickness was preserved. In rats that received saline at week 6 (**b**), optic nerve head excavation and reduction in optic nerve thickness was more pronounced. A similar effect was noted with all BPC 157 regimens. A total of 6 rats/group/interval.

## Data Availability

The data presented in this study are available on request from the corresponding author.

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
