# Peer review of "Stable Gastric Pentadecapeptide BPC 157 Therapy of Rat Glaucoma"

_biomedicines, 2021, doi:10.3390/biomedicines10010089_

Round 1
Reviewer 1 Report
- The introduction should be shortened, re-written focusing BPC 157 therapy and ocular vascular disease.
- Adding arrows indicating important findings in each picture in figures is desired.
Author Response
Reviewer 1
- The introduction should be shortened, re-written focusing BPC 157 therapy and ocular vascular disease
Introduction is considerably shortened, and focus on BPC 157 ocular therapy fully emphasized.
- Adding arrows indicating important findings in each picture in figures is desired.
The arrows were added to indicate important findings in each picture in figures.
Reviewer 2 Report
The revised text is significantly improved (a non-tracked version is needed).
Author Response
Reviewer 2
The revised text is significantly improved (a non-tracked version is needed).
Acknowledged. We appreciate this reviewer’s comment.
This manuscript is a resubmission of an earlier submission. The following is a list of the peer review reports and author responses from that submission.
Round 1
Reviewer 1 Report
This study evaluated BPC 157 therapy on retinal and nerve head gross morphology in rats, as a potential treatment for glaucoma. The results show promising protection against retinal damage in rats treated with BPC 157.
-The manuscript needs to be re-written for clarity, as-is, it is very difficult to read.
-For all graphs, please specify the sample size included in the study, distinguishing between technical replicates and biological replicates.
-Please define BPC 157 at first mention in the abstract and introduction.
Author Response
Reviewer 1
This study evaluated BPC 157 therapy on retinal and nerve head gross morphology in rats, as a potential treatment for glaucoma. The results show promising protection against retinal damage in rats treated with BPC 157.
-The manuscript needs to be re-written for clarity, as-is, it is very difficult to read.
-For all graphs, please specify the sample size included in the study, distinguishing between technical replicates and biological replicates.
-Please define BPC 157 at first mention in the abstract and introduction
Answer: Acknowledged and corrected. Introduction is entirely rewritten, Discussion accordingly corrected. We hope that reviewer will find the revised manuscript to be more readable. Sample size is consistently specified in all graphs.
Reviewer 2 Report
Comments for authors
- The introduction is too long. Please reconsider the Introduction section focusing BPC 157 therapy and ocular vascular disease.
- In Figure 2, adding arrows indicating important findings in each picture is desired.
- Please make the font same in Discussion section.
Author Response
Reviewer 2
Comments for authors
- The introduction is too long. Please reconsider the Introduction section focusing BPC 157 therapy and ocular vascular disease.
- In Figure 2, adding arrows indicating important findings in each picture is desired.
- Please make the font same in Discussion section.
Acknowledged and corrected. We appreciated the comment considering the Introduction section, and therefore, it is rewritten, and focused in accordance with the reviewer’s suggestion. It seems to us that Reviewer 2 was considering the Figure 3 (not Figure 2 as he/she mentioned), and therefore, in all Figures arrows were added, and the Figures improved (scale bars included). The font in Discussion is corrected.

Round 2
Reviewer 1 Report
The text should be revised to correct for all grammatical issues. The paper is incoherent. Please introduce the topic of study, provide a little background for the study, and then discuss the findings.
Examples:
First sentence of the abstract - "As highlight for the proposed glaucoma therapy arising from the rats with permanent major vessel 15 occlusion, the stable gastric pentadecapeptide BPC 157 therapy resulted with rapid activation and 16 upgrading of the collateral pathways shown to be effective against several severe syndromes 17 induced by permanent occlusion of major vessels, vein and/or artery, peripherally and centrally."
-First two sentences of the introduction are similarly not understandable and provide no context for the current study.
"In those with thrombotic glaucoma the intraocular pressure and its postural change 38 in the affected eye never decreased [1]. Thereby, we used the cauterization of three epis-39 cleral veins in rats as irremovable injury to approach possible alternative solution in the 40 leaving intact only one vein of the four episcleral rat veins [2,3]. If not upgraded, one 41 episcleral vein is regularly unable to acquire whole function, and means...''